# Facial Synthesis of Adsorbent from Hemicelluloses for Cr(VI) Adsorption

**DOI:** 10.3390/molecules26051443

**Published:** 2021-03-07

**Authors:** Yi Wei, Wei Chen, Chuanfu Liu, Huihui Wang

**Affiliations:** State Key Laboratory of Pulp and Paper Engineering, South China University of Technology, Guangzhou 510640, China; wygift@163.com (Y.W.); geogeo_chen@163.com (W.C.)

**Keywords:** hemicelluloses, hydrothermal carbonization, adsorption kinetics, heavy metal

## Abstract

It is challenging work to develop a low-cost, efficient, and environmentally friendly Cr(VI) adsorbent for waste water treatment. In this paper, we used hemicelluloses from chemical fiber factory waste as the raw material, and prepared two kinds of carbon materials by the green hydrothermal method as adsorbent for Cr(VI). The results showed that hemicelluloses hydrothermally treated with citric acid (HTC) presented spherical shapes, and hemicelluloses hydrothermally treated with ammonia solution (HTC-NH_2_) provided spongy structures. The adsorption capacity of the samples can be obtained by the Langmuir model, and the adsorption kinetics could be described by the pseudo-second-order model at pH 1.0. The maximum adsorption capacity of HTC-NH_2_ in the Langmuir model is 74.60 mg/g, much higher than that of HTC (61.25 mg/g). The green hydrothermal treatment of biomass with ammonia solution will provide a simple and feasible way to prepare adsorbent for Cr(VI) in waste water treatment.

## 1. Introduction

Nowadays, climate change and environmental pollution have raised increasing concerns, especially industrial waste water, which usually contains heavy metal particles and organic dyes [1,2]. Chromium, which been widely applied in manufacturing industries (such as dyeing, tanning, printing, polishing, and electroplating), is a typical heavy metal element in industrial waste water [3,4,5,6]. Chromium usually exists in various oxidation states, e.g., trivalent chromium and hexavalent chromium. Trivalent chromium is an important element in regulating glucose metabolism and maintaining normal tolerance in vivo. It can also affect the body’s lipid metabolism and reduce the contents of cholesterol and triglyceride in blood. However, Cr(VI) shows toxicity and carcinogenicity for creatures. Besides, Cr(VI) usually exists in the form of HCrO_4_^−^ and CrO_4_^2−^ in the environment, which has a high dissolubility in water. The high dissolubility of Cr(VI) in water allowed it to easily migrate into the digestive system and cause injures to human bodies [7]. National Health Commission (NHC, China) and China Environmental Protection Law have set the allowable limit (0.05 mg/L and 0.5 mg/L) for hexavalent chromium in human drinking water and industrial waste standards, respectively. It is necessary to prevent humans from harms caused by hexavalent chromium from various industrial waste waters [8,9].

To remove Cr(VI) from water, many methods have been developed, such as redox, biological treatment, adsorption, electrochemical processes, and so on [8,10]. Among these methods, adsorption is considered to be the most promising because of its simplicity, low cost, and low energy consumption. There are many kinds of adsorbents used for Cr(VI) adsorption in waste water, such as metallic oxide, polymer materials, and carbon-based adsorbents [11,12,13,14]. Owing to the controllable morphology, high surface area, low toxicity, low cost, and environmental-friendliness, carbon materials have become a kind of widely accepted adsorbent [15,16,17,18].

Owing to the simplicity, low cost, and environmental-friendliness, hydrothermal carbonization is one of the most promising methods for the production of valuable carbon materials, such as carbon spheres and porous carbons [19,20]. Hydrothermal carbons could be prepared from fossils, polysaccharides, and agricultural and forestry wastes, among others [21]. Noteworthily, agricultural and forestry wastes with extremely low cost, great availability, and relatively high carbon content [22,23] have become the proper candidate for producing carbon materials. For example, it was reported corn stalk and Tamarix ramosissima can be converted to lignite-like solid products after hydrothermal treatment, and heating values increased significantly [22]. Besides, carbon materials obtained by hydrothermal treatment usually have rich functional groups and excellent physical properties, as well as different morphology [24,25].

Although many works in the literature have reported different carbon materials used for Cr(VI) adsorption, these materials cannot achieve the desired adsorption performances (e.g., high removal efficiency, high adsorption capacity, without second pollution) [26,27]. Large numbers of adsorption experiments have shown that the adsorption capacity of carbon materials is highly dependent on pH, and the addition of the element nitrogen can also increase the adsorption capacity for Cr(VI). For example, polyethylenimine (PEI) grafted graphene oxide nanosheets were used for adsorption Cr(VI), and the optimum adsorption could be achieved at pH 2.0 and the maximum adsorption capacity was up to 1185 mg/g, which is the highest adsorption capacity [28]. This is related to the fact that amino (NH_4_^+^) is a kind of functional group with positive charges in adsorbents, and the element N can also provide electrons to promote the reduction of Cr(VI) [29].

In this study, hemicelluloses collected from chemical fiber company were used to prepare carbon materials with different morphology by hydrothermal treatment. The physical and chemical properties of samples were characterized by Fourier transform-infrared spectro-scope (FT-IR), X-ray diffraction (XRD), scanning electron microscopy (SEM), Raman spectroscopy, X-ray photoelectron spectroscope (XPS), and Brunauer–Emmett–Teller surface area analyzer (BET). We found the hydrothermal carbons could effectively remove Cr(VI) in Cr-containing aqueous solution. The results will help to develop economical and feasible adsorbents for environmental remediation.

## 2. Materials and Methods

### 2.1. Materials

Hemicelluloses were collected from a chemical fiber factory (Guangzhou, China) and dried for 2 days to constant quality before use. Potassium dichromate, anhydrous ethanol, and ammonia solution (AR) were purchased from Guangdong Guanghua Sci-Tech Co. Ltd. (Guangzhou, China). Citric acid monohydrate (AR) was purchased from Shanghai Lingfeng Chemical Reagent Co. Ltd. (Shanghai, China). Diphenyl carbamide was purchased from Shanghai Richjoint Chemical Reagents Co. Ltd. (Shanghai, China).

### 2.2. Carbon Materials Synthesis from Hemicelluloses

Two kinds of carbon materials were prepared from hemicelluloses by the hydrothermal method, and the production procedures were listed as follows.

Hemicelluloses (4 g) were dispersed in 60 mL of 0.1 mol/L citric acid solution. The mixture was transferred to a 100 mL reactor for hydrothermal reaction under the condition of 200 °C for 16 h in muffle furnace. After the hydrothermal treatment, the sample was washed three times with ultrapure water and ethanol (95%), respectively. The finally obtained carbon spheres, named hemicelluloses hydrothermally treated with citric acid (HTC), were air-dried in an oven at 110 °C for 12 h.

Hemicelluloses (4 g) were dispersed in 60 mL of (12.5%) ammonia solution and stirred with vigorous magnetic stirring for 3 h at room temperature, and then the hydrothermal step was started under the same conditions as HTC. The obtained product was washed three times with ultrapure water and ethanol (95%), respectively. The finally obtained products, named HTC-NH_2_, were dried at 110 °C for 12 h for further use.

### 2.3. Characterization

The samples were characterized by FT-IR (VERTEX 70, Bruker, Germany), BET (ASAP 2020, micromeritics company, Norcross, GA, USA), elemental analysis (vario EL cue, Elementar, Frankfurt, Germany), XPS (ESCALAB 250Xi, Thermo Fisher, Waltham, MA, USA), and SEM (Merlin, Zeiss, Jena, Germany). The adsorption test of samples was performed using ultraviolet–visible spectrophotometry (UV/Vis) (UV-1800, Shimadzu, Kyoto, Japan).

### 2.4. Adsorption Experiments

The adsorption capacity of the hydrothermal carbon materials was tested according to the previous literature [30]. The Cr(VI) solutions with different concentrations were prepared by diluting the stock solution (1 g/L). Typically, 0.1 g of sample was dispersed in 40 mL of Cr(VI) solution with different concentrations in a 150 mL conical flask. The adsorption experiment was performed in a water-bathing vibrator (Jintan Instrument Factory). After adsorption, the carbon materials were separated using a hydrophilic polytetrafluoroethylene PTFE syringe filter (0.22 um), and the concentration of Cr(VI) was detected with UV–Vis. The pH of the Cr(VI) solutions, ranging from 1.0 to 6.0, was adjusted with 1 mol/L HCl and 1 mol/L NaOH solution. The influence of adsorption parameters (including time (0–12 h), concentration of Cr(VI) solution (10–200 mg/L), and the dosage of adsorbent (0.5–10 g/L)) on the adsorption process was also investigated.

The adsorption capacity of the carbon materials was calculated according to the following equation:$$Q_{e}=\frac{\left(C_{0}-C_{e}\right)\;V}{m}$$
where *Q**_e_* (mg/g) is the adsorption capacity; *C*_0_ (mg/L) and *C**_e_* (mg/L) are the concentration of Cr(VI) in the solution before and after the adsorption experiments, respectively; V (L) is the volume of potassium dichromate solution in reaction; and m (g) is the dry weight of carbon materials.

## 3. Results

### 3.1. SEM and BET Analysis

The microstructures of HTC and HTC-NH_2_ were observed by SEM. As shown in Figure 1, the SEM images of HTC presented a homogeneous spherical structure with average diameter at 1–6 μm, while HTC-NH_2_ exhibited an irregular massive shape. The specific area of HTC is 3.06 m^2^/g, which is three times higher than that of raw hemicelluloses (Appendix A). In the sample HTC-NH_2_, there are lots of stacked particles with a uniform size of 30–50 nm dispersed on the surface of the sample, which led to the high specific surface area of HTC-NH_2_ (134.51 m^2^/g). The surface morphology of HTC-NH_2_ probably resulted from the lower degree of hydrolysis of the raw material due to the lack of H^+^, similar to the surface etched by KOH, and that might increase the active sites of HTC-NH_2_ for Cr(VI) adsorption [31].

### 3.2. FT-IR Analysis

FT-IR is a useful method to study the functional groups on the surface of samples. The FT-IR spectra of samples (HTC and HTC-NH_2_) are shown in Figure 2. According to the previous literature [32], the band around 3400 cm^−1^ corresponds to the –OH bending vibrations, while the band at 1638 cm^−1^ is attributed to the vibration of the –COOH and the C=O in the carbonyl, respectively. The existence of these peaks suggested that there were large numbers of carboxylic groups in HTC. In the spectrum of HTC-NH_2_, the absorption bands at 1430 and 1254 cm^−1^ are assigned to C–N and N–H groups [11], respectively, and the two peaks at 897 and 645 cm^−1^ are related to the out-of-plane N–H deformation vibration. These indicated the formation of chemical bonds between nitrogen and carbon atoms in HTC-NH_2_. Besides, the peaks at 2912 and 2990 cm^−1^ are attributed to the vibration of the C–H in the carbonyl, indicating the incomplete hydrolysis process in the ammonia aqueous solution.

### 3.3. XRD Analysis

The structural properties of HTC and HTC-NH_2_ were studied. In Figure 3, the XRD curves of HTC and HTC-NH_2_ both showed only one broad diffraction peak at about 20°, corresponding to the (002) inter-plane of graphite. This suggested the low degree of graphitization and the existence of abundant amorphous carbon in HTC and HTC-NH_2_ [33,34]. This could explain the obvious fluorescence interference, as presented in Appendix A, further confirming the low degree of crystallinity in the hydrothermal carbon materials.

### 3.4. XPS and Elements Analysis

Elemental analysis and XPS spectra could provide the elemental compositions and typical functional groups information of samples, as shown in Table 1 and Figure 4. As shown in Table 1, the content of nitrogen in the sample HTC-NH_2_ is higher than that in HTC. This indicated that nitrogen was dopped into HTC-NH_2_ during the hydrothermal process. In Figure 4b,c, the content of C–O/C–N in HTC-NH_2_ is also much higher than that in HTC, while the content of C=C/C–C in HTC-NH_2_ is much lower than that in HTC. These suggested the degree of carbonization of HTC-NH_2_ is lower than that of HTC. Figure 4d shows that there were some pyrrole and pyridine nitrogen in HTC-NH_2_, while pyrrole and pyridine nitrogen could not be distinguished from HTC. This resulted from some amino groups attached on the surface of HTC-NH_2_ during the hydrothermal carbonization reaction [35].

### 3.5. Effect of pH and Adsorbent Dosage

The initial pH of solution and the dosage of adsorbent could affect the adsorption capacity of carbon materials. Among these factors, it is particularly important to control pH of the adsorption process, because the pH of solution can exert influence on the form of Cr(VI) species and the surface charge of adsorption materials [36,37]. In this study, the Cr(VI) adsorption capacity affected by the different pH of the two carbon materials is shown in Figure 5. The initial Cr(VI) concentration was 50 mg/L, the dosage of adsorbent was 2.5 g/L, and the adsorption temperature was 298 K. Because the potassium dichromate solution exists in acidic form in nature, the pH range in adsorption of Cr(VI) was adjusted from 1.0 to 6.0 [29,38]. The results showed that the adsorption capacity of the two samples decreased with the pH increase, and the maximum adsorption amount was achieved when the pH reached 1.0 (HTC at 14.98 mg/g and HTC-NH_2_ at 17.77 mg/g), and almost decreased by 90% when the pH was up to 6 (0.40 mg/g for HTC and 1.76 mg/g for HTC-NH_2_), which is similar to the previous works [39,40,41]. According to the previous reports [29], Cr(VI) mainly exists in the form of CrO_4_^2−^ when pH > 6, and gradually changed to the form of HCrO_4_^−^ and Cr_2_O_7_^2−^ as the pH value decreased. Compared with HTC, HTC-NH_2_ displayed a higher adsorption capacity under the same pH. The higher adsorption capacity of HTC-NH_2_ was due to its larger specific area and amino groups (–NH_2_), which were the active adsorption sites for Cr(VI). When the solution is acidic, the functional groups –NH– and –NH_2_ could be protonated to –NH_2_^+^ and –NH_3_^+^, which leads to a strong affinity between the sample HTC-NH_2_ and Cr_2_O_7_^2−^ through electrostatic interaction, about 18% higher than the adsorption capacity of HTC [42].

The dosage of adsorbent in waste water can also affect the adsorption process [43]. In this work, the dosage gradient of adsorption materials was set as 0.5 g/L, and the adsorption temperature was 298 K. As shown in Figure 6, the removal rates of the two materials both increased with the increase of dosage. When the weight of adsorbent HTC-NH_2_ went from 0.5 g/L up to 5.0 g/L, 99.34% of Cr(VI) was removed, while the adsorption capacity of the adsorbent decreased by almost 60%. When the dosage of HTC increased to 7.5 g/L, 93.98% of Cr(VI) could be removed. Similar to HTC-NH_2_, the adsorption capacity of HTC also decreased to 6.37 mg/g. This is probably because more adsorption sites are exposed in solution, thus the increase of adsorbent dosage. However, the utilization rate for adsorption sites decreased with the increased adsorbent dosage, which led to the decreased adsorption capacity of the sample [44].

### 3.6. Adsorption Kinetics

The adsorption reaction will change with the time of contact between the adsorbent and the adsorbate solution until the adsorption reaches equilibrium. The kinetic experiments were carried out at the dosage of 2.5 g/L in a 50 mg/L Cr(VI) solution at 298 K, and the results are shown in Figure 7. The adsorption data were collected at 5, 10, 20, 40, 60, 120, 240, 480, and 720 min, respectively. The pseudo-first-order kinetic model and the pseudo-second-order kinetic model were used to fit the adsorption rate curves, respectively.

In the initial stage, there were more unoccupied active sites in the adsorbents, and the concentration of Cr(VI) in the solution was the highest, which resulted in adsorption quantity increasing rapidly. After the first 60 min, the Q_t_ reached 5.84 mg/g for HTC and 15.96 mg/g for HTC-NH_2_, accounting for 43% and 87% of the equilibrium adsorption capacity, respectively. This is because of the protonation of amino groups under acidic conditions making the adsorption easier for HTC-NH_2_. With the extension of the adsorption time, Q_t_ increased slowly until reaching adsorption equilibrium, and the time to reach equilibrium for HTC-NH_2_ was about 4.0 h, and that for HTC was about 8.0 h.

As shown in Table 2, the pseudo-second-order kinetics model with a higher correlation coefficients value (R^2^ = 0.96, 0.80) fitted better than the pseudo-first-order kinetics model (R^2^ = 0.89, 0.68), suggesting that the adsorption behavior of HTC-NH_2_ towards Cr(VI) corresponded to pseudo-second-order kinetics. These results indicated that the adsorption rate of Cr(VI) onto adsorbent was mainly determined by the chemical adsorption process, consistent with the previous results [45,46,47]. This is showed that the adsorption capacity of HTC-NH_2_ was up to 18.33 mg/g, which is much higher than that of HTC (13.68 mg/g).

### 3.7. Adsorption Isotherms

In order to gain insight into the surface properties and binding interactions between HTC-NH_2_ or HTC and Cr(VI), Langmuir and Freundlich isotherm models were utilized to analyze the adsorption isotherms of samples, The fitting curve was showed in Figure 8 and isothermal adsorption parameters was showed in Table 3. The adsorption behaviors of the adsorbent were studied at 298 K, 308 K, and 318 K, respectively. The initial pH was 1.0 and the adsorbent concentration of solution was 2.50 g/L. As shown in Table 3, the R^2^ values of the Langmuir model are slightly higher than those of the Freundlich model, which means the Langmuir model is more fitted to describe the adsorption of adsorbents in this work. There are abundant functional groups on the surface of adsorbents, which could be used as active sites for adsorption. These active sites can easily carry out electron transfer and present chemisorption in the adsorption process. Similar results have also been reported previously [48,49]. Besides, chemisorption is usually related to single-layer adsorption (Langmuir model) [50]. According to Section 3.6 adsorption kinetics analysis, the adsorbents in this work also present single-layer adsorption behaviors. Therefore, it is more reasonable to describe the adsorption behaviors of the adsorbents with the Langmuir model in this work. Based on the nonlinear Langmuir model, the maximum adsorption capacities of Cr(VI) onto HTC-NH_2_ and HTC were 74.60 mg/g and 61.25 mg/g, respectively. Compared with HTC, the adsorption capacity of HTC-NH_2_ was obviously higher, which was also related to the much larger specific area of HTC-NH_2_ (134.51 m^2^/g). This meant that the physical adsorption mainly depending on the specific area and chemical adsorption depending on the functional groups on the surface of adsorbent both played important roles in the Cr(VI) removal process. Compared with some hydrothermal carbon materials as reported in Table 4, HTC-NH_2_ showed excellent adsorption capability, and is more economical in hydrothermal additives.

## 4. Conclusions

In this work, two hydrothermal carbon materials (HTC-NH_2_ and HTC) from hemicelluloses were synthesized and used for Cr(VI) adsorption in solution. HTC has a good spherical morphology with a specific surface area of 0.97 m^2^/g. HTC-NH_2_ is an amorphous carbon with a spongy structure that has good pore structure with a specific surface area of 134.51 m^2^/g. The adsorption isotherms indicated that the adsorption capacities of HTC and HTC-NH_2_ for Cr(VI) were obtained from Langmuir model to be 74.60 mg/g and 61.25 mg/g, respectively. The adsorption processes of the samples were described by the pseudo-second order equation at pH 1.0. Compared with HTC, HTC-NH_2_ showed a higher adsorption capability, which was related to the relatively high surface area and the amino groups (-NH_2_). This facile route developed here may offer a possibility for producing economical biomass-based carbon materials for waste water treatment and great promises for industrialization.

## Figures and Tables

**Figure 1 molecules-26-01443-f001:**
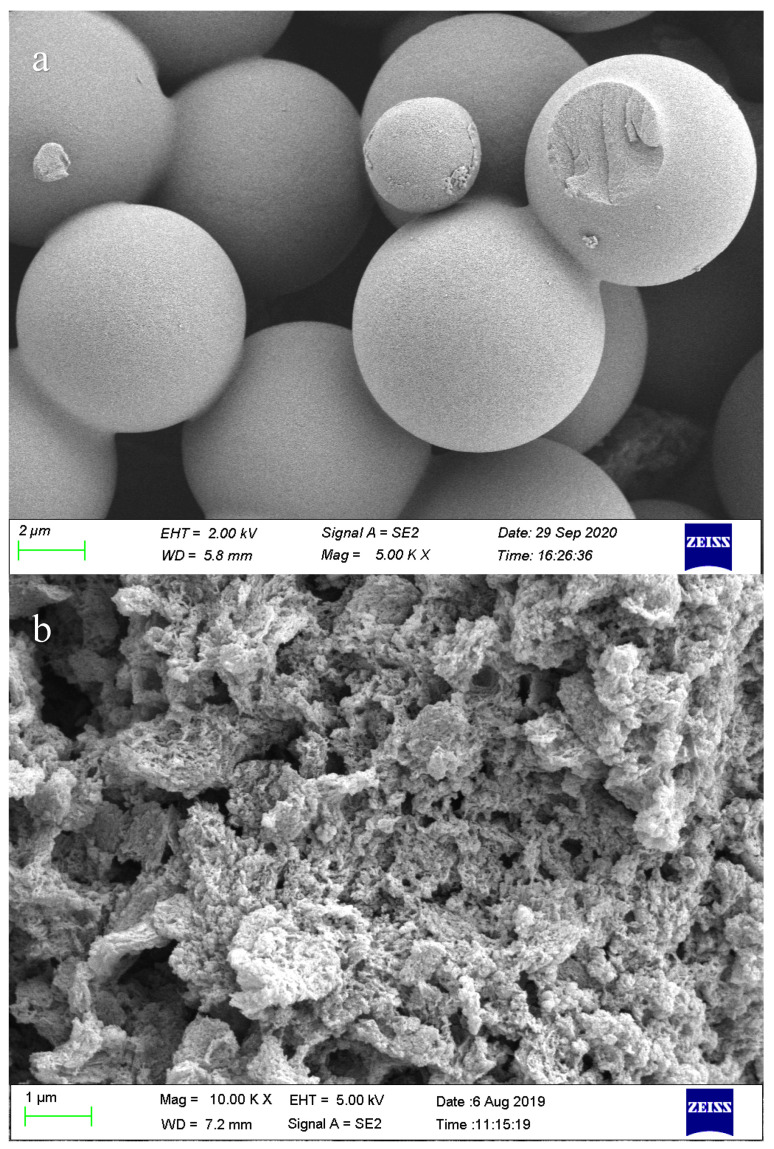
Scanning electron microscopy (SEM) images of samples: hemicelluloses hydrothermally treated with citric acid (HTC) (**a**) and HTC-NH_2_ (**b**).

**Figure 2 molecules-26-01443-f002:**
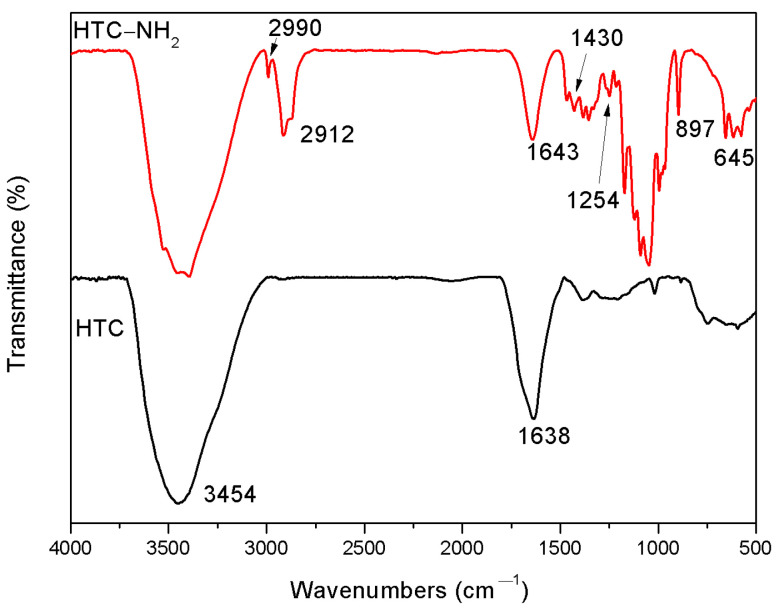
Fourier transform-infrared spectro-scope (FT-IR) spectra of HTC and HTC-NH_2_.

**Figure 3 molecules-26-01443-f003:**
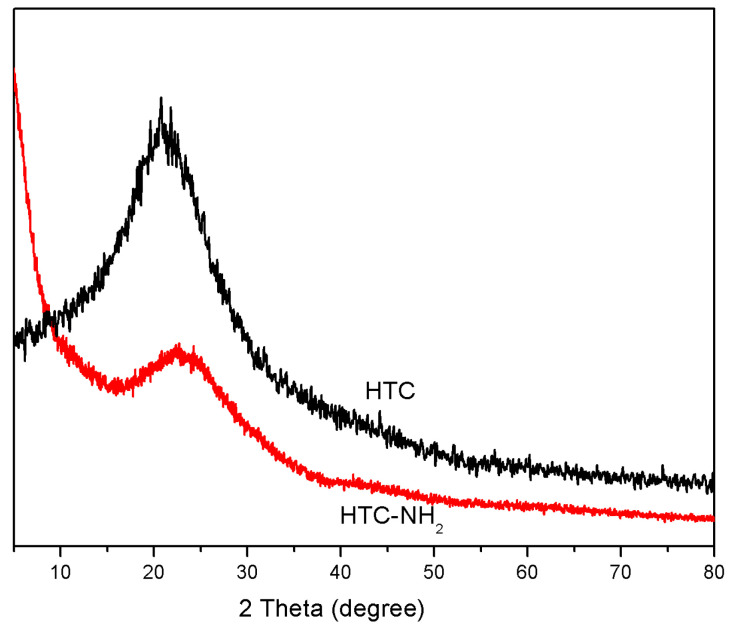
X-ray diffraction (XRD) curves of HTC and HTC-NH_2_.

**Figure 4 molecules-26-01443-f004:**
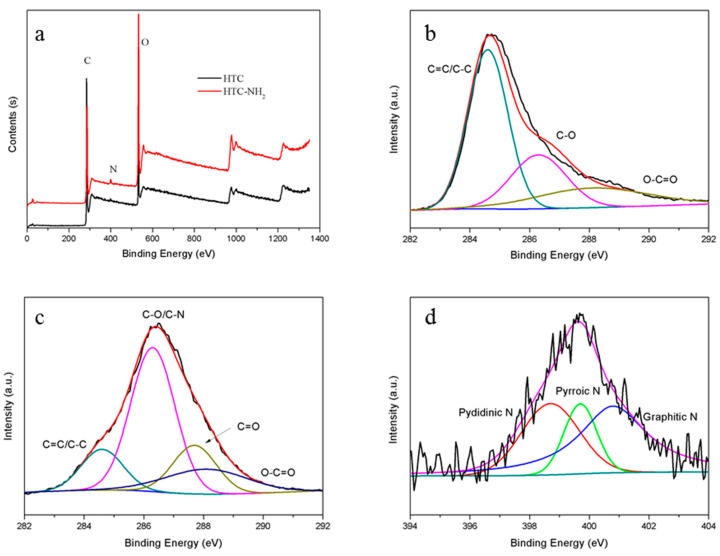
X-ray photoelectron spectroscope (XPS) spectra: (**a**) HTC and HTC-NH_2_; (**b**) C 1s spectra of HTC; (**c**) C 1s spectra of HTC-NH_2_; and (**d**) N 1s spectra of HTC-NH_2_.

**Figure 5 molecules-26-01443-f005:**
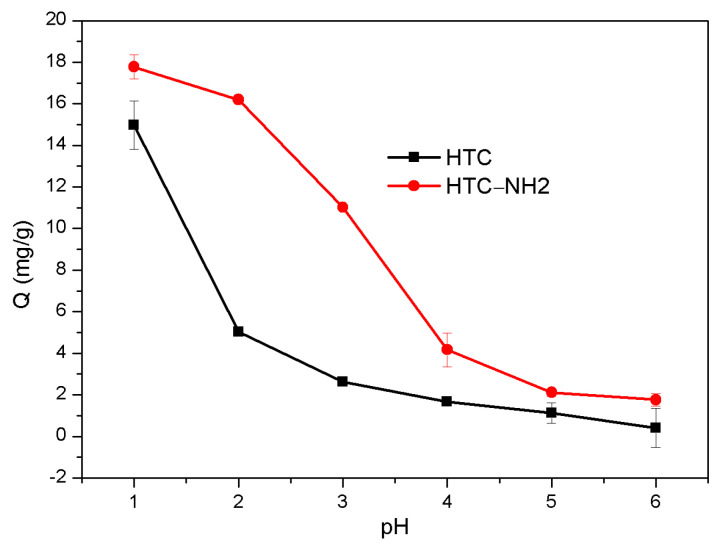
Effect of initial pH on Cr(VI) adsorption.

**Figure 6 molecules-26-01443-f006:**
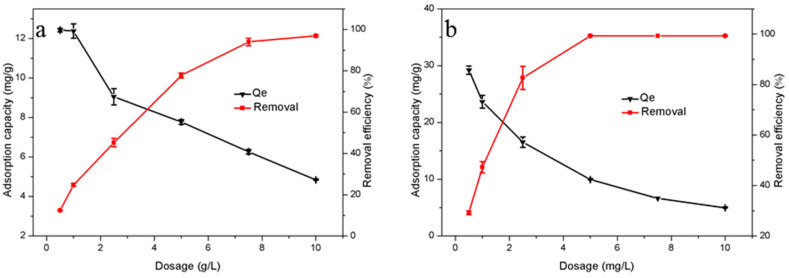
Effect of adsorbent dosage on removal of Cr(VI): (**a**) HTC and (**b**) HTC-NH_2_.

**Figure 7 molecules-26-01443-f007:**
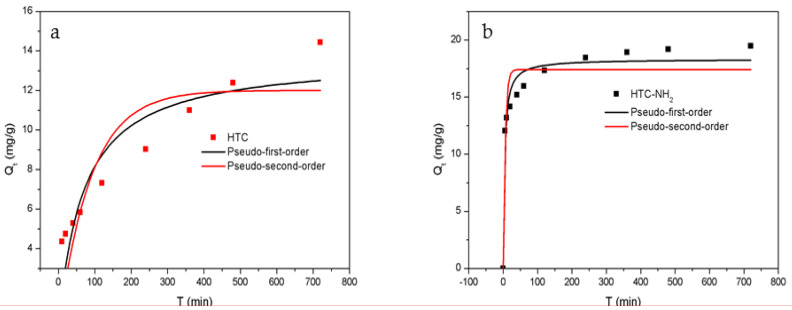
Adsorption kinetics of Cr(VI) onto samples: (**a**) HTC and (**b**) HTC-NH_2_.

**Figure 8 molecules-26-01443-f008:**
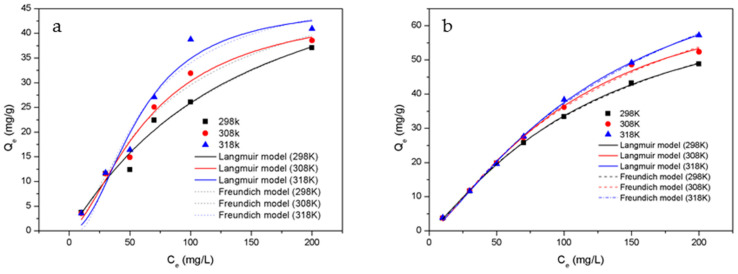
Adsorption isotherm fitting diagram of Cr(VI) onto samples: (**a**) HTC and (**b**) HTC-NH_2_.

**Table 1 molecules-26-01443-t001:** Elements contents of hemicelluloses hydrothermally treated with citric acid (HTC) and HTC-NH_2_ by elemental analysis.

Sample	Elements Content (%)					AAsh (%)
	C	H	O	N	S	
**HTC**	63.59	4.45	31.71	0	0.253	0.22
**HTC-NH_2_**	44.35	6.38	47.71	1.35	0.216	0.43

**Table 2 molecules-26-01443-t002:** Kinetic parameters for the adsorption process of Cr(VI).

Sample	Pseudo-First-Order			Pseudo-Second-Order		
	q_e_ (mg/g)	k_1_ (min^−1^)	R^2^	q_e_ (mg/g)	k_2_ (g/(mg min^−1^))	R^2^
HTC	12.01	0.011	0.68	13.68	67.82	0.80
HTC-NH_2_	17.41	0.18	0.89	18.33	3.74	0.96

**Table 3 molecules-26-01443-t003:** Isothermal adsorption parameters for Cr(VI) adsorption of samples.

Samples	Langmuir Model			Freundlich Model		
	q_max_ (mg/g)	b (L/mg)	R^2^	k	1/n	R^2^
HTC (298 K)	61.25	0.0051	0.9597	90.37	0.71	0.9588
HTC (308 K)	46.19	0.0014	0.9668	60.37	0.96	0.9563
HTC (318 K)	46.68	0.0002	0.9417	51.21	1.40	0.9176
HTC-NH_2_ (298 K)	74.60	0.0030	0.9995	110.18	0.77	0.9992
HTC-NH_2_ (308 K)	75.38	0.0018	0.9960	116.17	0.80	0.9937
HTC-NH_2_ (318 K)	88.57	0.0017	0.9989	148.75	0.75	0.9978

**Table 4 molecules-26-01443-t004:** Comparison of adsorption capacities with similar carbon materials.

Adsorbent	Treatment	pH/T (°C)	q_max_ (mg/g)	Ref
Salix hydrochar	hydrothermal	1/20	48.3	[51]
Magnetic biochar	pyrolysis	1/25	27.2	[41]
Nano-magnetite modified biochar	microwave treatment	3/25	26.7	[16]
Tectona grandis tree sawdust biochar	pyrolysis	3/30	83.5	[52]
Amino-functionalized magnetic biochar	Hydrothermal	2/25	142.86	[53]
biochar modified with nitric acid and nicotinamide	Hydrothermal	2/25	132.74	[54]
HTC	Hydrothermal	1/25	61.25	This study
HTC-NH_2_	Hydrothermal	1/25	74.60	This study

## Data Availability

All data included in this study are available in this published article and Appendix A.

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
