# Peer review of "Facial Synthesis of Adsorbent from Hemicelluloses for Cr(VI) Adsorption"

_molecules, 2021, doi:10.3390/molecules26051443_

Round 1

Reviewer 1 Report

1. In the abstract section, line 11 - adsorbent rather than absorbent should be written.

2. In the FTIR section, the authors did not provide very important information regarding the peaks appearing between 897-1254 cm-1 in the HTC-NH2 material. Also, there should be information about the y axis If its absorbance or transmittance.

3. In the table 2 the authors should provide information about the amount of Hydrogen present in the material and the ash content.

4. According to authors isotherm with the best fit was the Langmuir isotherm but studying table 4 it can be seen that Freundlich correlation coefficient is as high as Langmuir one. Additionally in the highest temperature parameter 1/n is higher than 1 suggesting that sorption is relatively hard for a sorbent to adsorb the ions. why ?

5. In the SI units there shouldn't be a space before percents and degree values. (line 214, Celsius)

6. Authors could add citation of the work: https://doi.org/10.3390/ma13122782

Author Response

Response to Reviewer 1 Comments

Point 1: In the abstract section, line 11 - adsorbent rather than absorbent should be written.

Response 1: Thanks for the reviewer’s comment. The word “absorbent” in the manuscript has been corrected as “adsorbent”, and colored red. We also polished the language in the manuscript with the help of researchers from University of Wisconsin-Madison.

Point 2: In the FTIR section, the authors did not provide very important information regarding the peaks appearing between 897-1254 cm-1 in the HTC-NH2 material. Also, there should be information about the y axis If its absorbance or transmittance.

Response 2: Thanks for the reviewer’s comment. The peaks between 897-1254 cm-1 are related with the characteristic peaks of hemicelluloses, which indicates the uncomplete carbonization of hemicelluloses during the hydrothermal process. The y aixs in Figure 2 is transmittance, which has also been supplemented in the manuscript. The FT-IR spectra of the raw hemicelluloses has also supplemented in the Figure A2 (Supporting Information).

Point 3: 3. In the table 2 the authors should provide information about the amount of Hydrogen present in the material and the ash content.

Response 3: Thanks for the reviewer’s comment. In this work, element contents on the surface of hydrothermal carbon materials were obtained by XPS analysis, which could not determine the element hydrogen and helium. Therefore, the element contents of samples were directly determined with elemental analysis, and the results were present in Table 1. The elements contents obtained from XPS analysis was supplemented in Table A2 (supporting information). The ash content of HTC and HTC-NH2 were 0.22% and 0.42%, respectively. This has also been supplemented in Table 1.

Point 4: According to authors isotherm with the best fit was the Langmuir isotherm but studying table 4 it can be seen that Freundlich correlation coefficient is as high as Langmuir one. Additionally in the highest temperature parameter 1/n is higher than 1 suggesting that sorption is relatively hard for a sorbent to adsorb the ions. why?

Response 4: Thanks for the reviewer’s comment. As shown in Table 4, the R2 values of Langmuir model is slightly higher than Freundlich model, which means Langmuir model is more fitted to describe the adsorption of adsorbents in this work. There are abundant oxygen-containing functional groups on the surface of HTC, which could be used as active sites for adsorption. These active sites are easy to carry out electron transfer and present chemisorption in the adsorption process. Similar results have also been reported previously [1,2]. Besides, chemisorption is usually related to single-layer adsorption (Langmuir model) [3]. According to section 2.5 adsorption kinetics analysis, the adsorbents in this work also present single-layer adsorption behaviors. Therefore, it is more reasonable to describe the adsorption behaviors of the adsorbents with Langmuir model in this work.

Point 5: In the SI units there shouldn't be a space before percents and degree values. (line 214, Celsius)

Response 5: Thanks for the reviewer’s careful and detail comments. We have eliminated the space between degree values in the manuscript and colored red.

Point 6: Authors could add citation of the work: https://doi.org/10.3390/ma13122782

Response 6: Thanks for the reviewer’s comment. We have added this citation in the manuscript, as shown in ref 46.

  1. Liu, L.; Cai, W.; Dang, C.; Han, B.; Chen, Y.; Yi, R.; Fan, J.; Zhou, J.; Wei, J. One-step vapor-phase assisted hydrothermal synthesis of functionalized carbons: Effects of surface groups on their physicochemical properties and adsorption performance for Cr(VI). Applied Surface Science 2020, 528, doi:10.1016/j.apsusc.2020.146984.
  2. Alatalo, S.M.; Repo, E.; Makila, E.; Salonen, J.; Vakkilainen, E.; Sillanpaa, M. Adsorption behavior of hydrothermally treated municipal sludge & pulp and paper industry sludge. Bioresour Technol 2013, 147, 71-76, doi:10.1016/j.biortech.2013.08.034.
  3. Kuleyin, A.; Aydin, F. Removal of reactive textile dyes (Remazol Brillant Blue R and Remazol Yellow) by surfactant-modified natural zeolite. Environmental Progress & Sustainable Energy 2011, 30, 141-151, doi:10.1002/ep.10454.

Reviewer 2 Report

This article evaluated two adsorbents made from hemicellulose for removal of Cr(VI) from wastewater. Meaningful results have been shown, while some concerns and problems are found. Please see below for my comments.

General comments:

The article is like a laboratory report. The authors described the experiments and showed the results. No in-depth discussion about the results was made. The authors need to give more detailed discussion associated with relevant mechanisms. When showing the background, the authors should specify the research gap and the objectives of this study. Previous work this study was based on needs to be reviewed and referenced. When demonstrating the results, suggestion should be made to indicate how the proposed materials could be potentially used in practice.

For all figures, when applicable, please show error bars to represent standard deviations of replicated trials. All tests should be done in replicate.

Specific comments:

Line 8: This is not clear. Why is it challenging?

Line 11: Absorbent or adsorbent?

Line 15: Do you mean the adsorbent should work at pH 1.0? Please clarify this. Is it practical at a so low pH?

Line 16: Such numbers should be obtained from replicated tests. Please show the standard deviations.

Line 17, “… will provide an simple and feasible way…”: This conclusion should be made based on a comparison with other traditional adsorbents.

Line 34: Which country do you refer to for NHC?

Line 35: I think you tested the materials for wastewater treatment. Please show the allowable limit of Cr (VI) in wastewater.

Lines 42 to 44, “Due to the controllable morphology…”: Provide more quantitative information with a comparison to other materials to support this argument.

Line 45: Explain why it is promising.

Lines 48 to 52: Show quantitative information and compare to other materials.

Line 54: Specify the desired adsorption performance.

Lines 55 to 57: Show quantitative results.

Lines 57 to 59: Can you show reactions to better explain the mechanism?

Lines 60 to 67: You stated that previous materials were not good. In this paragraph, you indicated you tested specific hemicelluloses. You missed an important demonstration in between these two statements - explaining theoretically how the methods you proposed could potentially solve previous drawbacks, which drove you to conduct this study. I think your method was built on other established methods. So please cite relevant papers in your demonstration. Were the materials you tested invented by yourself? If not, how did previous studies investigated and reported their performance? You need to have a literature review and make necessary discussion.

Fig 1: Captions under the photos are vague.

Table 1: You did not mention Table 1 in the text.

Line 104: Where is Figure A1?

Line 135, “Results showed that the adsorption capacity of the two samples decreased with the pH increase”: How much did the capacity decrease as a percentage?

Lines 136 to 138, “According to the previous reports…”: Since Cr(VI) salt changes its form at different pH, we need to investigate other situations with pH higher than 6. Please make a statistical analysis to indicate the normal pH range of wastewater with Cr(VI).

Line 139: Quantitatively show the higher adsorption capacity.

Lines 142 to 143: Quantifying how strong the affinity is, as compared to amino groups.

Fig 5: I think you need to use HTC instead of HTCA in the legend.

Lines 152 to 153, “However, the adsorption capacity…”: The adsorption efficacy also depends on the reaction time. Please show the information.

Lines 164 to 166: Indicate which model gives a better prediction.

Lines 170 to 172: Please make a quantitative discussion and explain why.

Line 177: Explain why second-order kinetics means chemical adsorption.

Lines 187 to 190: They have very close R2 values. HTC(308K) simulated by Freundlich model even has a higher R2 value. Why did you say Langmuir model was better?

Line 197, “…both played important roles…”: Can you discern their relative contributions to Cr(VI) as percentages?

Section 3: This section should be moved before results.

Lines 254 to 258: To justify the potential use of HTC-NH2, you need to compare its performance to other commonly used adsorbents and carry out a cost assessment of its operation.

Round 2

Reviewer 2 Report

Based on the authors’ responses and revisions made in the article, most of my comments have been addressed, while some have not. Please see the following for details.

I do not see authors’ responses to my general questions. Please address them. I do not see the section for material and methods in the revised version. You need to have this section.

Response 3: Please indicate whether it is practical at a so low pH value (pH = 1.0).

Response 5: Based on the comparison according to Table 4, please describe why this method is simple and feasible.

Response 7: It is confusing. Based on what you wrote, 0.05 mg/L is for drinking water treatment. Since you used industrial wastewater, you need to show the limit in wastewater.

Response 8: You indicate “controllable morphology, high surface area, low toxicity, low cost”. You need to show the numbers to justify this statement. For example, how low the cost is.

Response 14: You have not addressed my questions completely. Has this kind of material and method been investigated before? Please discuss what similar work reported.

Response 15: I mean the descriptive words just under the photos are too small to read.

Response 25: Please address my comments. When making this statement, make it as quantitative as possible. For example, quantitatively indicate the more obvious influence, the much shorter time, etc.

Response to 27: Please add this explanation to the article.

Response to 29: You cannot move this whole section to the supporting information. This section is important. You need to show it before Results.
